# ACCURATE COMPILER, OPTIMIZATION, AND ARCHITECTURE INDEPENDENT FUNCTION IDENTIFICATION USING PROGRAM STATE TRANSFORMATIONS

## Abstract

Patching vulnerabilities in third party libraries is critical for maintaining security, yet such patches can take over 500 days to be distributed on average. Manually creating binary patches requires semantic analysis to identify the full set of functions present in the library. Existing semantic binary analysis approaches do not scale, or are inaccurate. In this paper, we introduce IOVec Function Identification (IOVFI), which assesses similarity based on *program state transformations*, which compilers largely guarantee even across compilation environments and architectures. IOVFI executes functions with initial predetermined program states, measures the resulting program state changes, and uses the sets of input and output state vectors as unique semantic fingerprints. Since IOVFI relies on state vectors, and not code measurements, it withstands broad changes in compiler, optimization, underlying architecture, and even different implementations of equivalent functionality. Crucially, IOVFI is the first approach to support architecture independent classification.

Evaluating our IOVFI implementation as a semantic function identifier for `coreutils-8.32`, we achieve a high .779 average F-Score, indicating high precision *and* recall. When identifying functions generated from differing compilation environments, IOVFI achieves a 101% accuracy improvement over the most-recent *BinDiff 6*, outperforms `asm2vec` in cross-compilation environment accuracy, and, when compared to dynamic frameworks, *BLEX* and *IMF-SIM*, IOVFI is 25%–53% more accurate. Additionally, we show that IOVFI is largely unaffected by code obfuscation by achieving similarly high accuracy against obfuscated code. To demonstrate that state transformations are capable of cross-ISA identification, IOVFI achieves similarly high accuracy rates when identifying `AArch64` functions using unmodified `x64` classification vectors. We show that IOVFI scales to large binaries by evaluating semantic identification accuracy for three large and commonly used libraries: `libxml2`, `libpng`, and `libz`. Finally, we perform a semantic history analysis of `libpng` and `libz` on 14 different versions. We correctly identify `libpng` versions distributed with the last five years of Ubuntu releases.

## 1 Introduction

Semantic binary analysis—the act of determining a function's "purpose" within a binary—has applications in many research and engineering areas, such as plagiarism detection [48], code debloating [65], and malware analysis [8, 10, 34, 62]. Patching third party libraries [16, 23, 24, 49] requires determining the unpatched library version, *and* the full set of included functions, because developers will frequently distribute a custom-tailored version of a third party library that utilizes a subset of the possible functionality (e.g., only video decoding, and not encoding). [3] showed the first requirement is feasible to satisfy, but the second requirement is much less straightforward. Without source or an exact knowledge of how the library was generated, any user of a vulnerable library must either wait for the developer to fix the library, which can take on average over 500 days [3], or use semantic binary analysis to identify and locate vulnerable functions.

While source-based semantic inference work exists [28, 35, 36, 39, 44], semantic binary analysis is a more difficult problem [50] due to the lack of information at the binary level. Manual semantic analysis does not scale to large binaries, necessitating an automated solution. So far, automated binary analysis [18, 25, 33, 67, 72, 75] measures *binary code properties* (e.g., order and type of instructions [72], memory locations accessed [25, 45], or control flow [58]), and approximates semantic similarity of functions based on the similarity of code. Although it is true that code similarity implies semantic similarity, the converse is not true—machine code may vary while still preserving semantics. We demonstrate that *program state modifications* serve as a better, more *stable semantic function identifier*. Program state change as a function identifier relies on the fact that semantic behavior is stable across compilations, environments, and implementations. Thus, program state change provides an ideal fingerprint, as it is impervious to compilation environment diversity or information loss. Code measurement approaches are susceptible to these complicating factors.

We present IOVec Function Identification (IOVFI), an approach to precise binary semantic analysis. Instead of relying on fragile function code properties, IOVFI abstracts functions into characteristic sets of inputs and corresponding program state changes. The core idea of IOVFI is to observe and identify the *behavior or character* of functions instead of the underlying code, and then use the observed behavior as a unique function identifier. Our proof of concept IOVFI implementation automatically discovers a subset of a function's unique set of valid input program states and corresponding program state changes (referred to as Input/Output Vectors, or

*IOVecs*). By observing data flow and program state transformations, IOVFI can classify functions, and, as a first-in-class feature, the IOVecs can transfer to different architectures with minimal effort.

We evaluate our prototype on accuracy amid varying compilation environments, a task existing works find difficult yet is crucial for binary patching and reverse engineering. We measure accuracy by identifying functions in the `coreutils-8.32` application suite, and find that IOVFI achieves a high .779 average accuracy across 8 different compilation environments. When identifying functions from differing compilation environments, IOVFI is 101% more accurate than the static *BinDiff 6* [75] framework, and 25%–53% more accurate than the dynamic *BLEX* [25] and *IMF-SIM* [72] frameworks. IOVFI achieves similar results to asm2vec [22] when compilation environments are similar, and significantly outperforms it for differing compilation environments.

We further demonstrate the generality of IOVecs by achieving similar accuracy when analyzing obfuscated binaries and `AArch64` binaries using unmodified `x64` IOVecs. We also demonstrate that IOVFI scales to large binaries by analyzing `libxml2`, `libpng`, and `libz`, which shows only a linear growth in training time relative to the number of functions in the binary. As an illustration of the utility of IOVFI, we perform a semantic analysis of 8 different versions of `libz`, and 6 different versions of `libpng`, and measure significant semantic differences which correspond to major changes to the underlying source. Finally, we use the `libpng` IOVecs to identify the versions distributed over the past 5 years of Ubuntu releases.

This paper provides the following contributions:

1. Design of IOVFI, a framework for semantic binary analysis that infers function semantics through program state changes;

2. A practical implementation of IOVFI that leverages coverage-guided, mutational greybox fuzzing to automatically infer program states and input structure layouts for functions;

3. We show the effectiveness of IOVFI through a thorough evaluation on `coreutils-8.32`, obfuscated and cross-architecture binaries using unmodified IOVecs, and large shared libraries. We also perform a semantic analysis of 8 different versions of `libz`, and 6 different versions of `libpng`, and use the `libpng` training data to identify 5 years of Ubuntu distributed versions.

## 2   Challenges and Assumptions

Here, we outline challenges for semantic function identification, and our assumptions when designing IOVFI.

### 2.1   Semantic Function Analysis

Reverse engineering a binary is a tedious task. While initial extraction of binary code and determining the size and location of functions is non-trivial [4, 33, 40, 60, 61, 63], semantic identification is the hardest, most time-consuming part of reverse engineering. The largest impediment to semantically recognizing known functions is the large code diversity due to different compilation environments. Here, we refer to the compilation environment as the exact compiler and linker brand and version, optimization level, compile- and link-time flags, linker scripts, underlying source, and libraries used to generate a binary. Compilers attempt to create efficient, optimized code, and different compilers utilize different optimization sets. While compilers preserve the high level semantics expressed at the source level, the generated binary code is highly variable. For example, an analysis we performed on the `strlen` implementation in `musl` C library [53]—one of the simplest non-trivial functions in the C library—showed that simply changing the compiler could result in more than a 70% change in the disassembly. Optimizations, like dead code analysis and tail call insertions, also greatly affect the generated machine code. Even worse, custom function implementations (as opposed to the use of system-distributed libraries) will likely produce significantly different binaries.

However, regardless of compilation environment, the program state changes a function performs *must* remain stable for a binary to exhibit correct behavior. Barring any bug in the compiler implementation or inconsequential actions such as dead stores, the same source code should produce the same *semantic* behavior in the final application. If this was not the case, binaries would exhibit different, and likely incorrect, behavior in different builds. Therefore, measuring program state changes presents a viable method for semantic identification that does not rely on fragile measurements of code.

### 2.2   Assumptions

In line with existing semantic analysis tools, when designing IOVFI, we assumed the following:

1. Binary code is stripped, but not packed.

2. Binary code is generated from a high-level language with functions, and function boundaries are known.

3. Functions make state changes that are externally visible.

4. The binary follows a discernible and consistent Application Binary Interface (ABI).

5. Functions do not rely on undefined behavior.

When analyzing binaries, reverse engineers start with a stripped binary from which they infer its behavior. The analysts have no access to the underlying source, debugging

information, symbol table, or any other human-identifiable information. We assume the same setting for IOVFI. Semantic analysis frameworks also make the assumption that all code is unpacked, and that the binary was generated from a high-level language with a notion of individual functions and a known ABI. The latter assumption precludes applications written wholly in assembly with no discernible functions, and, while packed code is another serious challenge in binary analysis [21, 40, 63], that topic is orthogonal to the analysis that semantic analysis frameworks perform. Finally, as it is rare in practice and most likely a bug, no code may rely on undefined behavior to correctly function. The compiler is free to *use* undefined behavior for optimization purposes, but the original source should not rely on any specific compiler-based optimization utilizing undefined behavior for proper functionality. Note that functions which rely on randomness (e.g., cryptographic functions) are still valid; semantic analysis frameworks simply assume that function semantics do not change with the compiler.

## 3 IOVFI Design

IOVFI is a function identification framework, which infers program semantics by measuring the effects of execution. Instead of measuring code properties, it measures program state changes that result from executing a function with a specific initial program state. When a function executes, it does so with registers set to specific values, and an address space in a particular state, with virtual addresses mapped or unmapped to the process' address space, and mapped addresses holding concrete values. We refer to the immediate register values and address space state as the program state.

IOVFI performs its analysis by instantiating a specific program state before function execution, and then measures the program state post-execution. Measurable program state changes are writes to locations pointed to by pointers, data structures, and variables whose valid lifetimes do not end when the function returns. These types of changes necessarily must be made to registers or memory addresses outside the function's stack frame. This is because, once finished, any change would be overwritten by later instructions, and thus the program would have been more efficient had it not called the function at all. Functions that make only ephemeral changes are dead code, and the compiler will simply remove such code. Additionally, depending on optimization level, some program state operations, e.g., dead stores which write to addresses but are never read, can be removed from the final binary. We do not include such operations in the function's set of program state changes, but focus on *persistent* and *externally measurable* program state changes. We argue that most user space functions conform to these standards, however, we discuss the limitations these standards impose in § 6.

We also consider the immediate return value of a function to be a measurable program state change, but exclude

```
int my_div(int a, int b, int* c)
{    *c = a / b; return 0; }
```
Listing 1: An IOVec Motivating Example.

changes to general purpose registers (e.g., `rbx` on `x64`) and state registers (e.g., `rsp`). They are excluded because, for caller-saved general purpose registers, their values are immediately irrelevant upon function return, and state registers have no bearing on function semantics. Additionally, measurable program state changes preclude modifications to kernel state not reported to user space.

While executing, a valid program state for one function might cause another function to fault, and the same function can perform arbitrarily different actions based on the program state upon invocation. Therefore, a function implicitly defines the input program states it *accepts*—states where the function can run and return without triggering a fatal fault—and the corresponding output program states based upon these input states. We call these accepting input and corresponding output program states Input/Output Vectors, or *IOVecs*. A function *A* is said to accept an IOVec *I* if *A* accepts the input program state from *I*, and the resulting state from executing *A* matches the expected program state from *I*. If either of these conditions do not hold, then *A* rejects *I*. See § 3.3 for the discussion of matching program states. Assuming functions make changes to input program states which are measurable post-execution, we can reframe semantic function identification. Precisely identifying a function can be seen as identifying the *complete* set of IOVecs which a function accepts. We call that set the *characteristic IOVec set* (*CIS*).

Consider the toy example in Listing 1. An accepting input program state is one that has the first argument set to any integer, the second argument set to any integer except 0, and the third argument set to any properly mapped memory address. The memory location pointed to by `c` can initially have any value. The corresponding output program state has the return value set to 0, and the memory location pointed to by `c` contains the value of `a/b`. An IOVec is a single concrete tuple of accepting input state and corresponding output state, and $CIS_{my\_div}$ is the full set of IOVecs `my_div` accepts. Note that only the first two arguments, the location pointed to by `c`, and the return value, are relevant, and that neither the full address space nor every register value are relevant.

Every function has a *CIS*, and we hypothesize that most functions have a unique (non-empty) *CIS*. A set of functions that share a *CIS* is called an *equivalence class*. For the sake of brevity, unless otherwise noted, when we refer to a function, we are actually referring to an equivalence class of functions with equal functionality.

In the general case, a function's *CIS* is unbounded. So for practical reasons, we attempt to find a subset of a function's *CIS*, which we call the *distinguishing characteristic IOVec*

*set*, or *DCIS*. A *DCIS* for function $f$, $DCIS_f$, consists entirely of IOVecs which $f$ accepts, and only $f$ accepts every member of $DCIS_f$. Another function, $g$, might accept a member of $DCIS_f$, but there is at least one IOVec $I \in DCIS_f$ which $g$ does not accept. IOVFI is used to identify a function foo in a binary by providing foo with IOVecs $I_j \in DCIS_f$. If foo accepts *all* $I_j$s, then we say that foo $\equiv f$.

IOVFI needs an oracle to provide IOVecs in order to semantically identify functions, but there is no definitive source of IOVecs. Our prototype was designed to be one such oracle, but other oracles can be devised. For example, IOVecs can be derived from unit tests or inferred from a specification. Symbolic execution [6, 11] or constraint tracking [59] could similarly be leveraged to create IOVecs.

The number of IOVecs IOVFI needs in order to be precise is highly dependent on the diversity and number of functions analyzed. The minimal theoretical number is equal to the number of functions being analyzed, because IOVFI needs at least one accepting IOVec to identify and distinguish a function. However, it is likely more IOVecs are needed to precisely distinguish functions, but the use of *differences* in semantic behavior for discrimination minimizes the number of required IOVecs. We currently focus only on accepting IOVecs for semantic identification, however using rejected IOVecs also provides valuable feedback. For example, if a function $g$ rejects only 1 IOVec in $DCIS_f$, this can be a signal that $g$ and $f$ are semantically related.

IOVFI performs its analysis in two phases: a coalescing phase and an identification phase. The coalescing phase, which only needs to be run once, is where functions are classified by IOVec acceptances and rejections, and ordered into a binary tree accordingly. The second phase is where unknown functions are semantically identified by providing the unknown functions with specific IOVecs from the binary tree, and traversing the tree according to IOVec acceptance.

**Coalescing Phase**   IOVFI starts its analysis by providing every function in its training set with every IOVec the oracle provides. This establishes a full ground truth of which IOVecs are accepted and rejected, ensuring that proper ordering can be achieved. Recall that an IOVec encodes both an input state and expected output state. When an IOVec is given to a function $f$, one of four results can occur:

1. The function receives a fatal signal (e.g., SIGSEGV), due to an improper input program state.

2. The function does not return before a specified timeout.

3. The function returns, but the final program state differs from the expected output program state.

4. The function returns, and the final program matches the expected output program state.

IOVecs that satisfy the last result are added to $DCIS_f$. As future work, we want to incorporate rejected IOVecs into the identification process, as rejected IOVecs classify the *rejected* semantics of this function.

The result of the coalescing is a proposed *DCIS* for every function in the training set, and the *DCIS* then fed to a decision tree generator. We use a decision tree generator (as opposed to another machine learning classifier) because, decision tree generators make classifications based on information gain, which is ideal for IOVec acceptance and rejection. The output decision tree contains IOVecs as interior nodes, and functions at leaves, and can be used for semantically identifying any number of functions later. As the tree is generated using *differences* in semantic behavior, it only grows linearly in the worst case. Every path from root to leaf encodes a minimal *DCIS* needed to distinguish one function from every other in the tree. If the same path in the decision tree maps to more than one function, then a potential equivalence class exists in the binary. The functions in the leaf are those for which the generated *DCIS* is insufficient to fully distinguish one function from another. This can be because the generated IOVecs cover the functionality poorly, or the functions are truly an equivalence class.

**Identification Phase**   Figure 2 shows the overview of the identification phase. To semantically identify functions, the analyst provides IOVFI with an unknown binary and the generated decision tree from the coalescing phase. Starting from the root of the decision tree, the IOVec is given to the unknown function. If the IOVec is accepted, the *true* branch in the decision tree is taken; otherwise, the *false* branch is taken. The unknown function is then tested against another IOVec depending on the path taken. When the path arrives at a leaf, the unknown function is tested against one more IOVec from the leaf function's *DCIS* for confirmation. Again, if the IOVec is accepted, then the function is given the label of the function at the leaf. If the unknown function gets to a leaf and remains unconfirmed, then the function is labeled as unknown.

The policy used for determining matching program states must remain constant for both phases. For IOVFI, we have implemented one such policy (see § 3.3), but others can be devised. The program state matching policy should take into consideration the memory model and features of the language in which the functions are written.

### 3.1   IOVec Discovery

IOVFI requires an oracle to generate IOVecs. Our prototype implements a coverage-guided mutational fuzzer [5, 13, 17, 30, 31, 32, 43, 47, 57, 66, 68, 73, 74] to infer IOVecs. Since we have no information about an unknown function's semantic behavior, the ideas behind feedback-guided mutational fuzzing are useful in discovering IOVecs. By rapidly feeding a function random inputs, and measuring the program

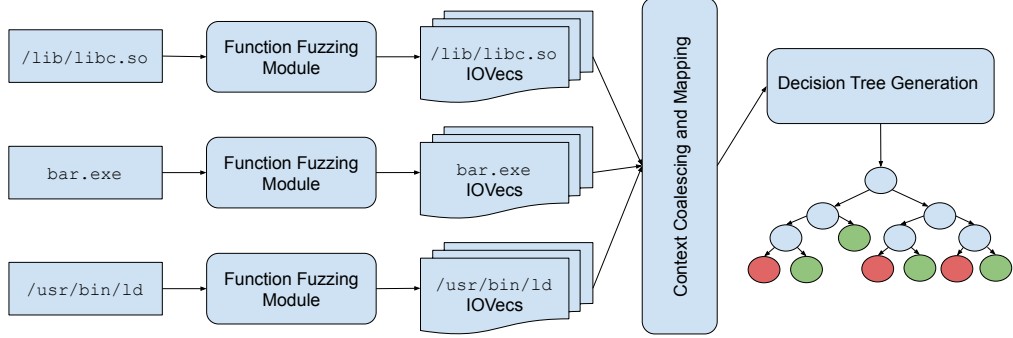

Figure 1: IOVFI Ahead-of-Time Learning Phase.

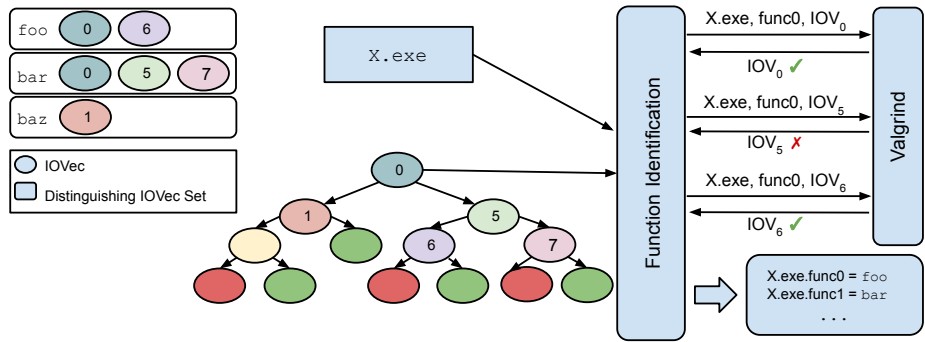

Figure 2: IOVFI Identification Phase. The ✓ and ✗ indicates that the IOVec was accepted and rejected respectively. Paths in the tree leading to green leaves indicate semantic equivalency in the unknown binary X.exe to a previously analyzed function (foo, bar, or baz), while paths leading to red leaves represent unseen/new behavior.

| IOVec Data | Use |
|---|---|
| Random seed | Program state initialization |
| Pointer input arguments | Program state initialization |
| Memory object information | Program state initialization |
| Code coverage | Fuzzer seed selection |
| Expected return value | Program state comparison |
| Expected memory state byte values | Program state comparison |
| Unique system calls | Program state comparison |
| Originating architecture | IOVec translation |

Figure 3: Data stored in IOVecs.

state change post-execution, we can build a corpus of function identification data without any *a priori* knowledge. We chose fuzzing as our exploration strategy because fuzzing is optimized to maximize code coverage, leading to maximal program state change coverage. We do not need full path or code coverage to be accurate, only enough program state change coverage (i.e., data-flow coverage) to differentiate semantics. While limitations of fuzzing (e.g., passing complex data checks [14]) may limit the quality of the IOVecs, we observe that they are sufficient in practice. Our experimental results reinforce our main claim that program state change (however the IOVecs are generated) provides a more stable semantic identification fingerprint than code measurements.

Figure 1 shows the overall design of the first phase of IOVec discovery and coalescing. Our prototype supports an-

alyzing any executable code, including shared libraries, but static libraries need to be included in either a shared library or executable. IOVFI requires neither the source nor any debug information; however, it does need boundary information of each function in an executable, or the exported symbol names in a shared library. Recent work shows that this information can be recovered even for stripped binaries [4, 60].

For each Function Under Test (FUT), our prototype fuzzes the input arguments and non-pointer memory object data if any have been deduced, and then begins executing the FUT with this randomized program state. If that program state is accepted, then the newly discovered IOVec is returned, and the resulting code coverage of the test is examined. If the IOVec produced new coverage, it is added to the FUT's *DCIS*, otherwise, it is discarded. Either way, the IOVec in the FUT's *DCIS* that produced the most coverage (or a completely new, randomized IOVec in case the *DCIS* is empty) is chosen as a seed for additional fuzzing. This process continues until the code coverage exceeds a user-defined threshold.

IOVFI stores the input program state and expected program state in an IOVec. Storing the entire address space is both a waste of storage and imprecise. Instead, IOVecs save the data listed in Figure 3. Memory object information is the coarse-grained input layout and global memory objects inferred during the generation of the IOVec, and includes location, size, and pointer sub-member offsets. While generating

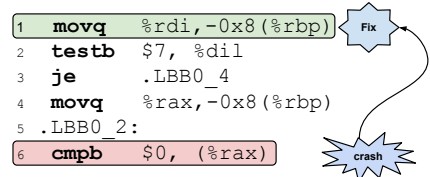

Figure 4: Backwards taint analysis to infer pointer arguments.

| Policy | Instruction | $t$ Taint? | $u$ Taint? | Taint Policy |
|--------|-------------|------------|------------|--------------|
| 1 | $t = u$ | Yes | No | T($u$); R($t$) |
| 2 | $t = u$ | No | Yes | |
| 3 | $t = u$ | Yes | Yes | |
| 4 | $t = t \circ u$ | Any | Any | |

Figure 5: Backwards Taint Propagation. $t$ and $u$ can be a register or memory address. T($x$) taints $x$ and R($x$) removes taint from $x$. $\circ$ denotes any logic or arithmetic operator.

IOVecs, our implementation uses code coverage to select an IOVec to mutate, so we include the instructions executed by the FUT when provided with the IOVec.

## 3.2 Pointer Derivation

A major challenge to generating high-quality IOVecs is the detection of pointers as input. As binaries contain no type information, determining if an input argument is a pointer is an ongoing research topic [51, 54]. Without recovering which arguments are pointers, determining a *DCIS* is generally impossible, and only incomplete behavior will be captured.

A simple solution would replace an invalid address with a valid address before an illegal dereference occurs. While such a solution has been successfully used to solve other problems in binary analysis [37, 56], it would not work in IOVFI, because the underlying problem—semantically, an input is supposed to be a pointer when it is not—remains. IOVFI relies on capturing program state changes that arise from executing a function with a specific input program state. By replacing an illegal address *in situ*, the resulting output state does not necessarily arise from actions performed given the initial state, and an IOVec with an input state and an unrelated output state would be generated.

Consider the code in Figure 4, which is adapted from the strlen implementation in musl. The first pointer argument (passed in using register rdi) is stored on the stack (line 1). Later, that address is written to register rax (line 4), and then is dereferenced and compared with the null terminator (line 6). Our fuzzing strategy is unlikely to supply a valid address as input, and line 6 will cause a SIGSEGV signal to be issued.

The simple approach would replace the invalid address in rax with a valid address. If the function later returns with no other issue, then IOVFI would register strlen as accepting the input program state with rdi set to a random (non-pointer) value. This is incorrect, and during the identification phase, an implementation of strlen in an unknown binary would *not*

accept the input program state. That strlen implementation would then be marked with an incorrect label.

The solution we propose is a backwards taint analysis inspired by Wang et al. [72], and illustrated in Figure 4. While generating IOVecs in its exploration phase, our prototype records immediate register values before every instruction executes, and, if a segmentation fault occurs (as in line 6 of Figure 4), we get the register containing the faulty address, which is the taint source. The saved register values are used to propagate the taint back to a root sink. The taint propagation policy is listed in Figure 5. Starting from the last executed instruction, each instruction is parsed in reverse order until all instructions are iterated through. The root sink is the last tainted register or memory address after all instructions are processed. Our implementation utilizes the *Valgrind* framework [54], and its architecture independent intermediate representation, VEX. As VEX instructions represent a single action, and we record all register values prior to executing a single machine instruction, we are able to precisely determine the root sink, and no false positives are possible.

After sink discovery, we search for previously allocated memory objects, and update the allocated bounds accordingly if an object is found. If no object is found near the faulting address, then a new memory object is built by allocating a fixed-size memory region, and records the current location and size of the object. We use this information for inferring new bounds and pointer sub-members if another segmentation fault occurs after execution restarts. Analysts can use the bounds information for more sophisticated analysis after decision tree generation. Once the object has been created or updated, the location is written to the sink, and begins executing the FUT from its beginning using the newly adjusted program state.

The backwards taint analysis restarts with every segmentation fault until the FUT successfully returns. When the FUT finally completes, we record the correctly initialized input program state, the corresponding output program state, and the coarse-grained object structure derived from the backwards taint analysis. IOVFI only tracks which memory areas are supposed to be pointers, and no other semantic meaning is given to memory regions containing non-pointer data. Further fuzzing iterations maintain the memory object structure, and only the non-pointer memory areas are fuzzed.

## 3.3 Matching Program States

IOVFI uses matching program states to differentiate and classify functions' semantics. Here, we present our definition of matching states used to identify C functions.

Recall that our notion of input program state includes memory objects for both global data as well as input arguments. Semantically similar functions modify memory objects in similar ways (if at all), so we capture the resulting memory state of allocated objects post-execution. Due to our fine-grained

control over the memory state, any pointer value (either as an input argument or as a structure sub-member) is the same across executions. The allocated memory objects can be any arbitrary data structure, containing a mix of pointer and non-pointer data at various locations within the structure. Program states match when non-pointer values in memory regions are byte-wise the same, and any pointers to sub-objects are located at the same offset from the object start. If there is a single mismatch in memory objects between two program states, then the states do not match.

Return values are also pertinent, but can be implementation dependent. We recognize two types of return values: pointers and non-pointers. Due to the lack of any type information in binaries, precisely determining if a return value is a pointer is challenging. We conservatively test if the return value maps to a readable region in memory, and if it does, we designate the return value as a pointer. If a return value is not readable in memory, then we consider it a non-pointer, and can represent functions that perform raw computations (e.g., `sin` or `toupper`), or adhere to a contract (e.g., `strcmp` which can return any value $< 0$, $= 0$, or $> 0$).

Finally, because system calls provide services that cannot be satisfied by user-space code and cannot be optimized out, semantically equivalent functions *must* invoke the same set of system calls. Order and number of system calls made, however, can differ among semantically equivalent functions (e.g., calling `read(fd, 1)` 4 times could be the same as calling `read(fd, 4)` once). Therefore, we include the set of unique system calls invoked while executing with the specific input as part of the IOVec. Semantically equivalent functions must invoke the same set of system calls, and can execute neither more nor fewer unique system calls.

For two program states to match, the values contained in return registers must match in the following ways. Return values must both be pointers or non-pointers. As we do not know the size of the underlying memory region, we do not check the underlying memory values if the return values are pointers; we simply say the return values match. Without more sophisticated analysis, this can be a source of inaccuracy. If the return values are non-pointers, they must be equal, or both must be positive or negative. If all input pointers (including pointers to all sub-objects) match, the return values match, and the same set of system calls are invoked, then the two program states match. As we do not perform any static analysis, `void` functions will also go through return value analysis, leading to another potential source of imprecision.

## 4 Evaluation

Our evaluation focuses on 64-bit System-V Linux binaries derived from C source code. We performed our evaluation using an Intel Core i7-6700K CPU, with 32 GB of RAM, running Ubuntu 16.04 LTS. We address the following research questions (RQ):

1. How accurate and scalable is IOVFI in identifying functions in binaries?

2. Is IOVFI truly resilient against compilation environment diversity?

3. Do IOVecs generated by IOVFI apply to other architectures?

4. Does IOVFI create meaningful equivalence classes?

Our results do in fact show that IOVFI is a feasible and accurate semantic function identifier. Additionally, our results show that IOVFI is largely unaffected by compilation environment changes, and that IOVFI can quickly identify previously analyzed functions. We show that IOVecs truly preserve semantics by achieving high accuracy when identifying functions in both purposefully obfuscated and `AArch64` binaries. Finally, our large-scale real-world application evaluation shows that IOVFI can scale to large, complex binaries.

### 4.1 Accuracy Experimental Setup

We selected `coreutils-8.32` for evaluation because the suite is a common evaluation metric in the literature, and used by both *BLEX* and *IMF-SIM* for their evaluation. To conduct our evaluation of IOVFI's accuracy, we selected `wc`, `realpath`, and `uniq`, which represent medium-sized applications using the default compilation environment. We compiled the set of applications using `gcc 7.5.0` [69] and `clang 6.0.0` [42], at `O0–O3` optimization levels. We then generated a decision tree (see § 3) for each application, for a total of 24 decision trees. The total amount of fuzzing time allocated for generating IOVecs was limited to 5 hours, after which the coalescing phase was allowed as much time as necessary. The coverage threshold to stop fuzzing a function was set at 80%. Only 19% of the classified functions hit that threshold during the exploration phase, and the average per-function coverage was 61%. While low coverage could miss important semantic features, Jiang et al. [36] found that in practice most functions are distinguishable using few executions. The accuracy in our evaluation further backs up this finding.

Each tree was used to identify functions in `du`, `dir`, `ls`, `ptx`, `sort`, `true`, `logname`, `whoami`, `uname`, and `dirname`, each also compiled using `gcc 7.5.0` and `clang 6.0.0` at `O0–O3` optimization levels, for a total of 80 binaries. These applications represent the 5 largest and smallest applications as determined by the default `coreutils` compilation environment. We used a subset of `coreutils` applications because an evaluation of one application requires $8 \cdot 24 = 192$ experiments. Evaluating all $100+$ applications would therefore exceed $20,000$ experiments. Given that the applications share a lot of functionality, such an exhaustive evaluation is unnecessary. In order to establish ground truth, we compiled all binaries with debug symbols enabled. However, IOVFI does

| D-Tree / Suite | | O0 | | | | O1 | | | | O2 | | | | O3 | | | |
|---|---|---|---|---|---|---|---|---|---|---|---|---|---|---|---|---|---|
| | | LLVM | | gcc | | LLVM | | gcc | | LLVM | | gcc | | LLVM | | gcc | |
| **O0** | LLVM | **.874** | *27* | .829 | *49* | .728 | *85* | .691 | *66* | .702 | *123* | .667 | *98* | .694 | *133* | .743 | *139* |
| | gcc | .852 | *56* | **.851** | *24* | .726 | *97* | .685 | *67* | .691 | *142* | .655 | *131* | .691 | *141* | .736 | *201* |
| **O1** | LLVM | .661 | *74* | .692 | *82* | **.891** | *30* | .636 | *66* | .753 | *79* | .690 | *73* | .718 | *73* | .671 | *110* |
| | gcc | .848 | *113* | .811 | *91* | .815 | *128* | **.852** | *34* | .808 | *137* | .782 | *104* | .804 | *146* | .854 | *146* |
| **O2** | LLVM | .723 | *107* | .744 | *121* | .836 | *76* | .736 | *89* | **.929** | *49* | .789 | *107* | .916 | *53* | .752 | *91* |
| | gcc | .710 | *85* | .757 | *117* | .835 | *117* | .718 | *74* | .828 | *129* | **.892** | *49* | .830 | *138* | .799 | *68* |
| **O3** | LLVM | .723 | *110* | .742 | *120* | .835 | *77* | .735 | *93* | .929 | *54* | .798 | *122* | **.926** | *51* | .760 | *99* |
| | gcc | .849 | *137* | .830 | *173* | .825 | *153* | .819 | *128* | .822 | *124* | .848 | *78* | .820 | *137* | **.932** | *53* |

Figure 6: Geometric mean F-Score (left) for `coreutils-8.32` per decision tree compilation environment (rows) across evaluation suite compilation environments (columns), and percent increase F-Score over *BinDiff 6* (right).

not use them for its analyses, and they were only used for determining accuracy after all analyses had completed. Unfortunately, some functions call `abort` or otherwise forcibly exit on invalid input, and thus our prototype in its current iteration could not properly analyze those functions.

We report the geometric mean F-Score (harmonic mean of precision and recall) across all compilation environments. In order to determine the correctness of a label, we performed a simple string comparison between the name of the FUT and the functions in the assigned equivalence class. If any matched, we record the function name as the assigned label, otherwise we use the name of the first function in the equivalence class as the assigned label. If a function is not matched to an equivalence class, we label the function as "*Unknown*". We then search for the function name among all the classified functions in the decision tree. The ground truth label is the function name if it appears in the classified function list, or "*Unknown*" otherwise. The classification labels and ground truth labels are then given to the `sklearn.metrics` Python module for F-Score calculation.

To evaluate against the most recent *BinDiff 6* (released in March 2020), we exported the needed input data using Ghidra [1] for each binary in every compilation environment, and performed pairwise analyses. The primary binaries were the decision tree binaries, and the rest of the binaries were the secondary binaries. Only the functions that our prototype classified were used for comparative accuracy measurements. We measured accuracy via a string comparison between matched function names, or with "*Unknown*" for secondary functions that cannot be matched. The primary matched name was considered as ground truth for matched functions. For secondary unmatched functions, the function name was used as ground truth if it was present in the primary function list, while "*Unknown*" was used otherwise. Unfortunately, *BinDiff 6* only provides one function name for matched functions, so no further analysis could be performed.

`asm2vec` [22] is another state-of-the-art static similarity framework that uses natural language processing to infer a model of functions using known function disassembly as training input. Function similarity is performed by computing the cosine difference between two numerical vectors derived from the trained model, where one vector represents a known function, and one vector represents the FUT. The pair that yields the highest cosine difference is assigned equivalence. `asm2vec` will always return a similarity score (and a match), even when presented with a function the model has not seen. This feature presents a challenge in fairly evaluating IOVFI against `asm2vec`, because IOVFI is capable of declaring the untrained function as "*Unknown*," while `asm2vec` can only return a value between $[-1, 1]$[1].

To evaluate against `asm2vec`, we trained a separate model using the binary tree binaries, and used each model, along with the binary's functions, to identify functions in the test set. The function names of the top 2 results were compared with the FUT name, and the FUT name was used as the label if there was a match, or the top result label was used if there was no match. We used the top 2 results to fairly compare against the average equivalence class size that IOVFI differentiates (see § 4.3). Unfortunately, due to the long evaluation time needed for `asm2vec` (see the discussion in § 4), we could not evaluate it using the full training and test binaries. Instead, our `asm2vec` evaluation consists of the O0 and O3 `clang` and `gcc` decision tree binaries, and `true` and `logname` as test binaries, again only using the O0 and O3 versions. `true` and `logname` were chosen as representative of the small and large binaries in our evaluation set. We report the average F-Score `asm2vec` achieves while varying the compilation environment, along with the average true label cosine similarity, and the average predicted label similarity. The high F-Score that `asm2vec` achieves when the test and training binaries match compilation environments shows that, while imperfect, this evaluation is reasonable given how `asm2vec` produces results. Due to the evaluation concerns with `asm2vec`, our evaluation focuses on *BinDiff 6*, as that system provides a more fair apples-to-apples comparison, despite its lower accuracy relative to `asm2vec`.

---

[1]Technically, 0 could be construed as "*Unknown*", but utilizing it would be challenging, as most functions have *some* similarity with each other on the assembly level, and thus a 0 cosine similarity is rare.

## 4.2 Accuracy Amid Environment Changes

Figure 6 shows the geometric mean F-Score IOVFI achieved with decision trees from a specific compilation environment, along with the percent increase over the geometric mean F-Score achieved by *BinDiff 6* with the same environment. Each row reports the accuracy of all decision trees or primary applications from the specific compilation environment has when used to identify functions in binaries generated with a specific compilation environment (presented as the columns). The diagonal numbers (in bold) are, therefore, the accuracy rates when the decision trees or primary applications and evaluation suite match in both compiler and optimization level. They are unsurprisingly among the most accurate IOVFI and *BinDiff 6* achieved, and represent the data most reported by related work. *BinDiff 6* achieved an overall $.402 \pm .111$ accuracy and standard deviation, and a diagonal accuracy of $.642 \pm .0387$. Overall, we achieve a $.779 \pm .0777$ accuracy rate, while the diagonal accuracy is $.893 \pm .0331$, an improvement of 39%.

The off-diagonal numbers represent situations where training binaries differ from the evaluation binaries, and highlight the limitations of *BinDiff* and the strengths of IOVFI. As a static analysis framework, *BinDiff* performs its analysis using various graph comparison and hashing heuristics. While static analysis is significantly faster, those heuristics are based on fragile properties, as instructions and control flows change with compilation environments. Conversely, IOVFI relies on program state changes, which compilers guarantee will be stable across compilation environments and, as we will demonstrate, even across architectures. *BinDiff 6* achieves an average off-diagonal F-Score of $.380 \pm .0726$ while our prototype achieves an off-diagonal $.766 \pm .0682$ average F-Score, an improvement of 101%.

The generally high F-Scores across compilation environments indicate that our accuracy largely comes from IOVFI's ability to identify functions it has classified, and not from simply assigning an unknown classification to functions it has not identified. These results show that IOVFI is accurate as a semantic function identifier, as well as resilient to compilation environments (RQ 1 and 2).

Figure 7 shows `asm2vec` accuracy and similarity scores. We achieve similar F-Score values when the compilation environments match those of the training binaries, however IOVFI significantly outperforms `asm2vec` when the compilation environments differ. We attribute our use of F-Score as the accuracy metric, and the tight restrictions on the predictions that `asm2vec` produces for the difference between our results and the existing literature. Restricting the results to the two highest similarity functions, and incorporating both precision and recall into the accuracy metric makes achieving high accuracy a strictly more difficult task. The authors of `asm2vec` show that their system exhibits a inverse relation between precision and recall, which our results confirm. Conversely, IOVFI achieves high precision and recall.

The middle and right values of Figure 7 list the average similarity scores measured for the true labels and for those that were picked as predictions respectively. In almost every case, the label similarity score is higher than the actual similarity score, indicating that incorrect functions are being measured as more similar than the correct function. Furthermore, the similarity scores measured for true labels from different compilation environments are significantly lower that those from matching compilation environments. For example, the true similarity score when using an `LLVM-O0` model to classify `LLVM-O0` binaries (0.973) is 45% higher than the scores measured while classifying `gcc-O0` binaries (0.537). As `asm2vec` claims, the further from 1.0 two vectors are, the less related the corresponding functions are, and, therefore, it was unexpected to measure such low (and even negative) average similarity among different compilation environments. The low similarity scores for the off-diagonal entries indicates that `asm2vec` is not well suited to analysis of binaries across varied compilation environments.

Additionally, we question the scalability of systems like `asm2vec` as semantic identifiers for large amounts of trained binaries. As noted earlier, our evaluation is only a subset of the full *BinDiff 6* evaluation because it could not complete in reasonable time. The main cause of the long processing time lies in the fact that the function vectors that `asm2vec` generate are independent entities that cannot be sorted in a meaningful way. Because of this independence, every unknown function must be tested against every classified function in order to provide sound results. Conversely, IOVFI's ability to sort IOVecs into a binary tree creates an $O(nlog(n))$ vs. $O(n^2)$ classification disparity that results in significantly reduced binary classification time. We measured an average single vector pair comparison time to be small, taking only $0.12 \pm 0.012$ CPU seconds on average across $3,223,276$ vector comparisons, which is inline with the published literature. However, when all pairs of classified and unclassified functions must be compared, the total aggregate time to classify an unknown binary becomes large. IOVFI takes significantly longer to train, with `asm2vec` taking only 4 CPU minutes to train a model from one binary, versus hours for IOVFI. However, we emphasize that the training only needs to be done once, and afterwards classification with IOVFI is quick (see § 4.4).

Unfortunately, the two closest dynamic systems to IOVFI, *BLEX* [25] and *IMF-SIM* [72], are not available publicly. The *BLEX* authors supplied us their code, but it required significant engineering to execute with currently distributed Python modules. We invested two weeks of development and evaluation time. The accuracy we measured was much lower than the reported values, but this could be attributed to the required engineering changes or changes in the imported modules. The *IMF-SIM* authors remained unresponsive. We, therefore, base our comparison with these dynamic works on the published numbers, and call for open-sourcing of research prototypes. The *BLEX* authors report an average accuracy of .50–.64

| Training \ Test | | O0 | | | | | | O3 | | | | | |
|---|---|---|---|---|---|---|---|---|---|---|---|---|---|
| | | LLVM | | | gcc | | | LLVM | | | gcc | | |
| O0 | LLVM | **.952** | **.973** | **.969** | .224 | .537 | .642 | .0379 | .270 | .497 | .0199 | .333 | .548 |
| | gcc | .296 | .596 | .704 | **.951** | **.966** | **.965** | .0379 | .291 | .500 | .0467 | .479 | .636 |
| O3 | LLVM | .0656 | -.218 | .535 | .0370 | .283 | .586 | **.849** | **.955** | **.949** | .159 | .612 | .626 |
| | gcc | .0519 | -.0407 | .453 | .0108 | .295 | .565 | .220 | .381 | .511 | **.857** | **.920** | **.939** |

Figure 7: `asm2vec` F-Scores (left), average similarity of true labels (middle), and average similarity of predicted label (right).

across three compilers (they added Intel's `icc` compiler) and four optimization levels, and the *IMF-SIM* authors report an average accuracy of .57–.66 across three compilers and three optimization levels. Both systems attempt to build a classification vector from code measurements, and their lowest accuracies come from labeling functions in binaries from compilation environments different from their source models. IOVFI, in contrast, is accurate regardless of compilation environment, as evidenced by the off-diagonal numbers in Figure 6. With a geometric mean accuracy of .766, our results show an average 25%–53% increase in accuracy in differing compilation environments over these works. The inaccuracy in *BLEX* and *IMF-SIM* arises from the fact that code measurements are not a true reflection of function semantics, but are instead one way to express function semantics from a large and diverse space of possible semantic expressions. The trained models they generate become inaccurate when presented differently optimized code, because they only capture a small portion of the possible semantic expression space. IOVFI achieves its accuracy by actually measuring a function's semantics through program state change, and does not approximate function semantics through code measurements.

Despite its higher accuracy, IOVFI does have inaccuracy. We identify two major sources of inaccuracy: an overly strict program state comparison, and kernel state dependence leading to low-quality IOVecs.

**Strict State Comparison**   In § 3.3, we detailed our policy for comparing program states, which we use in lieu of code measurements for determining semantic similarity. We opted for a strict policy where both return values and allocated memory areas must match exactly in order for an IOVec to be accepted. However, at lower optimization levels, we might capture dead stores that are optimized out at higher optimization levels. For example, the `c_isprint` function, which returns a single byte, contains an additional `movzx` instruction in `O0` not present in any later optimization level. This instruction operates on the return register, which changes the higher order bits, while higher optimization levels simply write to the lowest byte in the return register without changing any further bit value. The write to the higher order bits is a dead store, since any caller will only ever read the lowest byte of the return register. However, we capture this behavior in an IOVec, and our strict return value comparison policy determines the return values to be different, leading to a mislabel. This is not a fundamental flaw with IOVFI, but an artifact

| | O0 | | O1 | | O2 | | O3 | |
|---|---|---|---|---|---|---|---|---|
| | LLVM | gcc | LLVM | gcc | LLVM | gcc | LLVM | gcc |
| $N$ | 78 | 73 | 72 | 52 | 38 | 32 | 36 | 40 |
| $\overline{N}$ | 1.76 | 1.85 | 1.82 | 1.68 | 1.78 | 1.47 | 1.69 | 1.69 |

Figure 8: Geometric mean count of classified functions ($N$), average number of functions per equivalence class ($\overline{N}$) for all `coreutils-8.32` generated decision trees. The median equivalence class size is 1.00 for all decision trees.

of our program state matching policy. A different policy that more precisely compares program state could better account for inconsequential program state changes.

**Kernel State Dependence**   For simplicity, we designed IOVFI to assume nothing when generating IOVecs, and it always executes functions in isolation. However, there are functions (e.g., `close` and `munmap`) that depend on the results of previous functions in order for the input arguments to be semantically correct. For instance, `close` requires that the input integer be a valid open file descriptor (as obtained from `open`), and any input that is *not* a valid file descriptor is semantically incorrect. Because we do not perform any initial setup to obtain semantically correct input values, any IOVec generated for these functions only exercise the error checking functionality, which is likely to be similar to many other functions. This has two negative effects: unrelated functions get grouped into an equivalence class, and unrelated FUTs can be assigned to this equivalence class simply because they share similar error handling behavior. This is, again, not a fundamental flaw in IOVFI, but instead is a result of our focus on user-space functions. We expect that our accuracy would improve significantly if we added some common environmental activities (e.g., opening file descriptors or memory mapping address spaces) to our IOVec design. We keep it as future work to incorporate application specific environmental setup to IOVFI.

### 4.3   Equivalence Class Distributions

Figure 8 shows the geometric mean number of classified functions ($N$), and the average number of functions per equivalence class ($\overline{N}$). Ideally, $\overline{N}$ should be close to one, as most functions provide unique and singular functionality, and thus should be assigned as the sole member of an unique equivalence class. However, with the existence of wrapper functions,

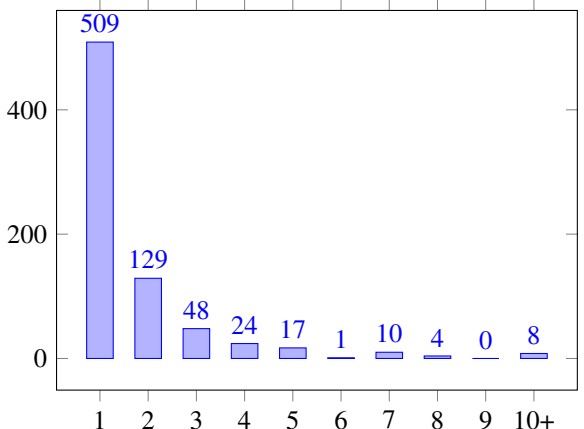

Figure 9: Distribution of all equivalence class sizes across all decision trees in the `coreutils-8.32` evaluation.

it is likely $\overline{N}$ will be higher. It nevertheless should be low, because one could trivially get high accuracy by grouping all functions into the same equivalence class. As Figure 8 shows, we achieve a low $\overline{N}$ across our decision trees, which indicates that our fuzzing strategy is a generally sound technique for generating sufficiently distinctive IOVecs. Additionally, the equivalence class size distributions in Figure 9 show that we are creating hundreds of equivalence classes with one or two functions per equivalence class, which provides evidence that we satisfy RQ 4. We, therefore, claim that our accuracy comes from IOVFI's ability to distinguish function semantics, and that our prototype does not simply group all functions into a few equivalence classes.

There are equivalence classes containing a large (10+) number of functions. These are cases where our fuzzing strategy was unable to trigger deep functionality, yet the classified functions share a common failure mode (e.g., return $-1$ for invalid input), or very similar functionality. For example, there is a 12 function sized equivalence class in the `realpath clang-O1` decision tree that contains 8 functions `strcaseeq[0-7]` that perform the same action with increasingly fewer input arguments. Improvements in related fuzzing work, especially works that improve deep code coverage [13, 14], will directly translate to an improvement of IOVec generation, and a reduction of the size of these equivalence classes.

## 4.4 Training and Labeling Time

IOVFI is scalable in both training time and storage requirements. On average, IOVFI takes 24.3 CPU hours to generate a decision tree, which includes generating IOVecs and the coalescing phase described in § 3. As stated before, however, this analysis only needs to be done one time. Once the decision tree is generated, semantic analysis is very quick, taking, on average, only 13.0 CPU minutes to classify a binary in the evaluation set. Additionally, all operations in both of IOVFI's phases represent completely independent work loads, and as

such are embarrassingly parallel. Therefore, execution time varies with the available hardware. Furthermore, the generated decision tree size is very small, with an geometric mean size of 855.9 KB. So, while IOVecs have no upper bound in their spatial size as they record the memory state of relevant inputs and their sub-members, in practice they are small.

*BLEX* reports $1,368$ CPU hours for training, and 30 CPU minutes to classify a binary in `coreutils`. *IMF-SIM* takes $1,027$ CPU hours for training, and 31 CPU minutes to classify a `coreutils` binary. Due to significant hardware differences between our respective experimental setups, and the lack of available source code for the related work, we cannot make any fair quantitative comparison. However, we believe that we are faster at semantic queries as we organize past analysis in a tree structure; *BLEX* and *IMF-SIM*, like `asm2vec`, must compare the feature vector they record with every past feature vector. Neither works report spatial size of their feature vectors, however *BLEX* and *IMF-SIM* restrict the number of instructions executed, which caps the size of their respective feature vectors.

## 5 Case Studies

We provide four case studies that demonstrate the effectiveness of our approach.

## 5.1 Accuracy Against Obfuscated Code

Malware authors will often employ code obfuscation to impede binary analysis [7, 9]. Code obfuscation attempts to hide semantic meaning through code transformations, such as adding unrelated control-flow or instruction substitution, while still preserving the intended function semantics. Code-based semantic analysis can be stymied when attempting to identify purposefully obfuscated code, because the resulting code is far from "normal," and thus hard to correlate with models derived from unobfuscated binaries. IOVFI, however, relies on *semantic* (rather than code) measurements guaranteed to be preserved by code obfuscators. Therefore, IOVFI should largely be unaffected by code obfuscation.

To test this hypothesis, we compiled our `coreutils` suite (`du`, `dir`, `ls`, `ptx`, `sort`, `true`, `logname`, `whoami`, `uname`, and `dirname`) using the LLVM-Obfuscator [38] at `O2`, enabling separately the bogus control-flow (bcf), control-flow flattening (fla), and instruction substitution (sub) obfuscations. Following the experimental methodology of the *IMF-SIM* authors, we used the `O0` decision trees to measure semantic function identification accuracy in each of the three respective obfuscated binaries, using the same accuracy measurement metric described in § 4.1.

The results are listed in Figure 10. We match or exceed the results achieved by *IMF-SIM*, with an average increase in accuracy of $39.3\%$. Our accuracy against obfuscated binaries, which closely matches our accuracy against unobfuscated

| | | IOVFI | *IMF-SIM* | % Difference |
|---|---|---|---|---|
| gcc | bcf | 0.787 | 0.385 | 105 |
| | fla | 0.772 | 0.576 | 34.1 |
| | sub | 0.752 | 0.664 | 13.2 |
| LLVM | bcf | 0.806 | 0.513 | 57.1 |
| | fla | 0.795 | 0.649 | 22.5 |
| | sub | 0.813 | 0.779 | 4.30 |

Figure 10: Obfuscated code accuracy comparison when bogus control-flow (bcf), control-flow flattening (fla), or instruction substitution (sub) is enabled for `coreutils-8.32`.

| | O0 | | O1 | | O2 | | O3 | |
|---|---|---|---|---|---|---|---|---|
| | LLVM | gcc | LLVM | gcc | LLVM | gcc | LLVM | gcc |
| 1 | .835 | .805 | .789 | .840 | .797 | .803 | .795 | .860 |
| 2 | .820 | .803 | .766 | .794 | .740 | .761 | .737 | .842 |
| 3 | .880 | .866 | .833 | .791 | .799 | .849 | .796 | .877 |

Figure 11: F-Scores for identifying functions in `coreutils-gcc-O3` AArch64 binaries using decision trees generated from x64 `wc` (1), `realpath` (2), and `uniq` (3).

binaries, provides evidence that IOVFI is unaffected by existing obfuscation techniques. Any inaccuracy when identifying functions in obfuscated binaries comes from the same sources as analyzing normal binaries, as discussed in § 4.2. Furthermore, these results also give evidence that RQ 2 is answered, as not only are the binaries purposefully obfuscated, but are also compiled using a much older version of LLVM than our evaluation version.

## 5.2 **AArch64** Evaluation

Function semantics are mainly determined by the high level source code, and remain largely constant across architectures. How the input state is established, and how the resulting program state is determined post-execution will change with architecture, but semantics do not. Therefore, an IOVec generated for one architecture is usable for another architecture, as long as there is a suitable IOVec translation between the two. In our implementation, we created a translation from x64 IOVecs to AArch64 IOVecs.

We evaluated IOVFI's cross-architecture accuracy by compiling the `du` and `dirname` (the largest and smallest binaries in our evaluation suite) on a Raspberry Pi 3 Model B Rev 1.2 running Ubuntu 20.04 using the ARM `gcc-9.3.0` compiler at `O3` optimization. We then used the unmodified decision trees generated for the evaluation described in § 4.1 to identify functions in the ARM binaries. The results are presented in Figure 11, with each column listing the accuracy achieved using the x64 decision tree generated with the enumerated compilation environment.

We achieve a mean F-Score of .811 across all the evaluated binaries, similar to our native geometric mean of .779. As our accuracy is largely unaffected by architecture, we strengthen our claim that IOVFI captures function *semantics*, and provide

| | O0 | | O1 | | O2 | | O3 | |
|---|---|---|---|---|---|---|---|---|
| | LLVM | gcc | LLVM | gcc | LLVM | gcc | LLVM | gcc |
| A | - | .871 | .717 | .850 | .759 | .746 | .765 | .772 |
| B | - | .781 | .633 | .695 | .629 | .642 | .629 | .639 |
| C | - | .794 | .699 | .802 | .701 | .722 | .700 | .733 |

Figure 12: F-Scores identifying functions in `libz` (A), `libpng` (B), and `libxml2` (C) using a `clang-O0` decision tree. We did not evaluate against the `clang-O0` binary.

| | `libz` | `libpng` | `libxml2` |
|---|---|---|---|
| $N$ | 126 | 390 | 2080 |
| $\overline{N}$ | 2.47 | 2.48 | 2.44 |
| $T$ | 17.0 | 25.4 | 158 |

Figure 13: Decision tree ($N$), average equivalence class sizes ($\overline{N}$), and CPU hours needed to generate the decision tree ($T$).

evidence that we answer RQ 3. Additionally, we also provide further evidence that we answer Research Question 1, as the `gcc` version used for this evaluation differs from the version used to generate the decision trees.

## 5.3 Large Shared Libraries

Here, we demonstrate the scalability of IOVFI to larger, more complex binaries.

We chose `zlib`, `libpng`, and `libxml2` as a set of shared libraries that are ubiquitous and among the largest distributed with Ubuntu. We compiled each library using `gcc 7.5.0` and `clang 6.0.0` at `O0–O3` optimization levels, generated a decision tree for the `clang-O0` binary, and identified functions in the remaining binaries. Due to the larger size of the binaries involved, we allowed the fuzzing campaign to execute for 10 hours, and provided as much time as needed for coalescing. In order to handle the significant increase in functions, we used a machine with 45GB memory to generate the decision tree for `libxml2` (running Debian 9.3 on an Intel Xeon 3106). The machine listed in § 4.1 was used for all other evaluation tasks. The 50% increase in memory to process at least a 10x increase function count is a reasonable cost, and does not detract from our scalability claim.

Figure 12 and Figure 13 list the accuracy measured (using the same accuracy metric at in § 4.1), along with the number of functions classified ($N$), average number of functions per equivalence class ($\overline{N}$), and CPU time required to generate the decision tree ($T$). Our prototype achieves similar F-Scores as in our `coreutils` evaluation, while showing only a linear growth in $T$, demonstrating the accuracy and scalability of our approach (RQ 1). However, the number of functions per equivalence class is higher than our `coreutils` evaluation. This is a consequence of our simplistic coverage-guided fuzzer, as well as increased genuine similar functionality. For example, there are functions in `zlib` (e.g., `gzoffset` and `gzoffset64`) which only differ in the bit count of their input arguments, but otherwise perform the same action. There are

also a large group of functions which first perform a sanity check on the input. The fuzzer did not create inputs to pass these checks, and the functions are grouped into an equivalence class. Although inferring valid input is an ongoing research topic [13, 14, 57], both of these problems can be mitigated with a longer fuzzing campaign, a more sophisticated fuzzer, or through symbolic execution.

## 5.4 Semantic Differences and Versioning

Semantic function identification is required for binary patching if the compilation environment that created the binary is unknown. A binary might contain only a subset of the functions available in the source code, and identifying the full set of functions allows an engineer to generate a patch for any vulnerable function. IOVFI, since it is unaffected by compilation environment, is well suited to identify and locate functions within a binary for patch generation.

To demonstrate IOVFI's utility in binary patching, we analyzed the latest 8 versions of the `zlib` compression library, spanning `1.2.7` to `1.2.11`, as well as 6 versions of `libpng` identified in the LibRARIAN [3] Android app dataset. We kept the default compilation environment (`gcc O3`) constant across all versions, generated decision trees for each resulting shared library, and then used each tree to identify functions in every other version. As in the `coreutils` evaluation, if the FUT name appeared in the assigned equivalence class, then we considered the two versions of the FUT to be semantically equivalent, and otherwise, the semantics differed. Additionally, we manually verified a subset of mismatched functions for code changes resulting in semantic differences.

The differences in function semantics as a proportion of classified functions is listed in Figure 14, along with the number of additions and removals to source files between each pairwise version as reported by `git`. While some versions show sharp differences in semantics, (e.g., `zlib v1.2.9+` is significantly different from earlier versions), subtle semantic differences are also distinguished. As IOVFI does not rely on any information, besides function location within a binary, and the majority of functions within both shared libraries are not exported, we claim that IOVFI can uniquely identify exported and non-exported functions.

Key benefits of IOVFI are low analysis time to construct the dataset and very low matching time to query a function. The full semantic difference analysis of all 56 `zlib` version pairs took only 82 CPU minutes, while the 30 `libpng` comparisons only took 54 CPU minutes. Other approaches must compare each unknown function with every generated function model, creating an $O(log(n))$ vs. $O(n)$ search performance disparity between IOVFI and the current state-of-the-art.

Figure 14 shows that binary versions often have measurable semantic differences from each other, and thus those differences can serve as an identifying fingerprint for a particular version. When analyzing the exported functions of a shared

library of an unknown version using decision trees generated from known library versions, the decision tree that produces the highest accuracy is likely to be the closest version to the unknown binary. LibRARIAN [3] performs this task statically (at a lost off precision), but IOVFI has the additional benefit of identifying non-exported symbols.

To test IOVFI as a shared library version identifier, we obtained the versions of `libpng` distributed for the past 5 years of Ubuntu releases, and analyzed each version with the `libpng` decision trees generated for the semantic difference evaluation. The decision tree with the highest accuracy was chosen as the candidate version, and we declared a successful match if that version is the closest to the actual version. The library versions include `1.6.37-3build3`, `1.6.37-3`, `1.6.37-2`, `1.6.37-1`, `1.6.36-6`, `1.6.34-2`, `1.6.34-1`, and `1.6.25-1`. In all but the `1.6.25-1` trial, IOVFI determined the correct version. For the unsuccessful trial, IOVFI selected the `1.6.37` decision tree, instead of the correct `1.6.24` decision tree.

## 6 Discussion

Here we provide discussion on the limitations of IOVFI, and on when a function is designated as unknown.

**Limitations** We have identified a few sets of functions that IOVFI is unlikely to classify or identify correctly. These functions are highly dependent upon the system environment and execution context while generating IOVecs, as well as during the identification phase. Functions like `getcwd` or `getuid`, which return the current working directory and the user ID respectively, depend on the filesystem, current user, and kernel state. As these factors differ between runs or are non-deterministic, they violate our fundamental assumption—semantically similar functions change their program state in similar ways given a specific input program state. To address this limitation, IOVFI could model the system state in addition to the process state.

Another set of functions IOVFI struggles with depend on an initial seed being set beforehand. Examples of these functions include `rand` and `time`. As we execute functions without any knowledge about their behavior, we cannot provide the seed beforehand as it is difficult to distinguish a seed value from other global variables. Even if we determine a location of the seed, knowledge of proper API usage (e.g., calling `srand` before `rand`) is needed to correctly use these functions. Discerning correct API usage is an active research area [2], and improvements in this area will directly translate to improvements in IOVFI.

**Soundness of IOVFI** When semantic equivalence is determined between two functions, that equivalence is only extended as far as the IOVecs tested along the decision tree path.

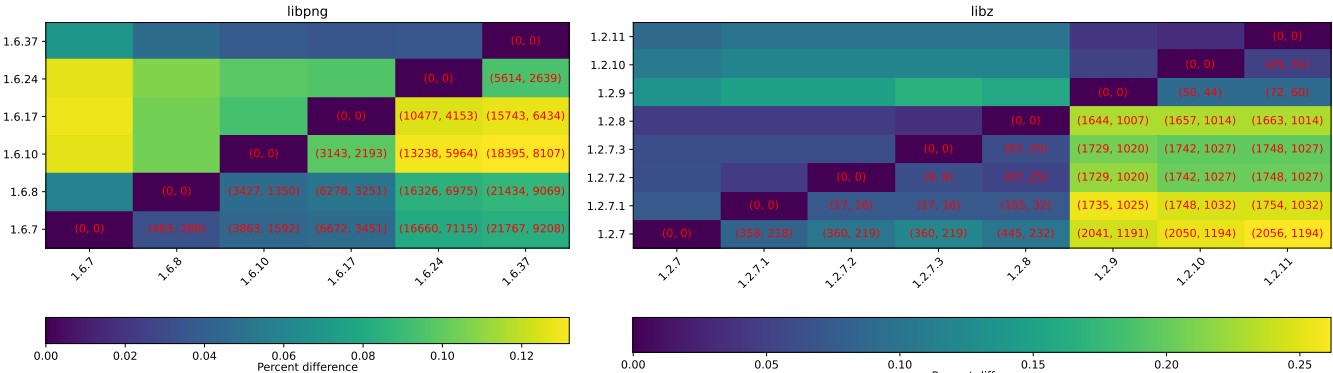

Figure 14: Semantic changes measured by function mismatches between IOVecs generated for a particular version (rows) and other versions (columns). Labels indicate the number of line additions and removals in source files between versions.

It is possible that IOVFI establishes an incorrect semantic equivalence between a previously analyzed function f, and a new unseen function g, if 1) g accepts all of f's IOVecs, plus additional IOVecs; and 2) any additionally accepted IOVec is not along the path to f in the decision tree. This means that IOVFI is not a sound technique. However, as our equivalence class distributions results show, in practice IOVFI is accurate for most functions, even when functions are similar, as with strcpy and strncpy. In real world code, most functions have little overlapping functionality, which makes IOVFI a practical tool for semantic identification. We leave it as future work to incorporate code coverage into the semantic similarity analysis, which could produce more accurate classifications through the enforcement of a coverage policy as a condition for semantic equivalence.

**System Calls**   Currently, IOVFI only records which system calls are made during the execution of a FUT, and no further information is captured, and no further modeling is performed. This design choice is purposefully incomplete to avoid expensive operating system state replication. Most functions make no system calls; less than 3% of functions in libxml2 call read, write, open, close, or their FILE* equivalents, for example. Of the functions that do make a system call, we assumed that they ignore the exact state of the operating system, and rely solely on the result of the system call. Our high accuracy justifies this assumption, and while state modeling could improve coverage, we believe that only marginal gains would result.

**Unknown Functions**   If a function is encountered that accepts no known *DCIS*, IOVFI will mark this function as unknown. When a function is marked as unknown, it can mean one of two things depending on the number of accepted IOVecs. If the unknown function *never* accepts an IOVec, then it implements wholly unknown functionality, and should be a main focus for analysts. Otherwise, if the function accepts some IOVecs, then it shares some functionality with the func-

tions whose *DCIS* includes the accepted IOVecs. The utility analysts might gain from this information varies with the number of IOVecs accepted. Many IOVecs rejected with a few IOVec acceptances is likely a common failure mode present in many functions, e.g., returning $-1$ on invalid input. If many IOVecs in a *DCIS* are accepted, then the unknown function is likely similar to the corresponding function, indicating, e.g., a different version.

# 7   Future Work

IOVFI utilizes a mutational fuzzer to generate a function's *DCIS*. By incorporating more sophisticated fuzzing and binary instrumentation techniques [15, 71], it is possible to generate a *DCIS* that provides close to 100% edge or code coverage of a function. Later, if that function is identified in a new binary, then any deviation in code coverage when given the full coverage *DCIS* would indicate the presence or lack of functionality in the FUT. This could be helpful in exploit generation, or code version identification [10].

A challenging aspect of reverse engineering is the detection of cryptographic functions in a binary. They are difficult to identify, because they are often implemented using architecture-specific assembly for optimization purposes, make extensive use of randomness, and rely heavily on correct state and input. These are situations for which IOVFI is particularly well-suited, and it would be worthwhile to investigate how far we can advance automated analysis on this most difficult class of functions. IOVecs, as an extension of the captured state, could record the random values returned by RNGs. In the coalescing and identification phases, calls to RNGs could be intercepted, and the recorded random value could be returned.

## 8  Related Work

Similarity analysis is an active area of research [12, 19, 20, 22, 26, 27, 29, 41, 46, 52, 64, 70]. Jiang et al. [36] first proposed using randomized testing in function similarity analysis, drawing inspiration from polynomial identity testing. Their EqMiner system, which requires source code, finds syntactically different yet semantically similar code fragments in large (100+ MLOC) code bases. A direct comparison between IOVFI and EqMiner is unfortunately challenging. Besides requiring source, which IOVFI does not use, the correctness metrics and similarity assertions used between the two systems are different. For example, EqMiner will declare two functions similar if they add two integers, irrespective of whether the integers are part of a struct or raw data types. IOVFI will mark the two functions as different, because of the different semantic uses in the whole binary. EqMiner defines similarity orthogonally to the data format while IOVFI uses IOVecs as the fundamental distinguishing factor. Both answers are correct for their respective use cases, but are incompatible when trying to evaluate one system over the other.

Current state-of-the-art binary analysis tools all rely on code measurements. *BLEX* [25] extracts feature vectors of function code, such as values read and written to the stack and heap, by guaranteeing that every instruction is executed. The authors also implemented a search engine with their system similar to IOVFI. Wang, et al. [72] perform code similarity analysis using a system called *IMF-SIM*. *IMF-SIM* uses an in-memory fuzzer to measure the same metrics as *BLEX*, instead of forcing execution to start at unexecuted instructions. As stated in our evaluation, these works still struggle with differing compilation environments, while IOVFI has consistently high accuracy irrespective of compilation environment. Both works focus on measuring code properties, which change with different compilation environments. IOVFI, in contrast, uses IOVecs, which are independent of code, and encodes differing semantics in a binary decision tree.

Pewny, et al. [58] compute a signature of a bug, and search for that signature in other (possibly different ISA) binaries. The signature involves computing inputs and corresponding outputs to basic blocks in functions' CFGs through dynamic instrumentation similar to IOVFI. While the authors admit that semantic function identification is not their expected use case, their system can be used as such by supplying a function as the "bug." This work relies on the structure of the CFGs of both the application's functions and the code being searched for, which can significantly change with software version or obfuscation. IOVFI is resilient to such differences as long as the function's semantics remain the same. Unfortunately, we were also unable to obtain source code or detailed results for comparison.

DyCLINK [70] use dynamic analysis to compute a dependency graph between instructions executed during developer supplied unit tests. Code similarity is determined by computing an isomorphism between sub graphs, using edit distance between PageRank [55] vectors. DyCLINK targets Java applications so we cannot compare our prototype against it. DyCLINK considers methods as similar if they share *any* sufficiently similar behavior for a given input, an event much more prevalent in C binaries than Java binaries. Many dissimilar C functions behave similarly when handling errors (i.e., returning −1 on invalid input), while Java often favors raising different exceptions based on the error condition. We, therefore, believe that the common error handling technique in C would significantly affect DyCLINK's precision. IOVFI is able to distinguish between functions with similar functionality, because the decision tree, which encodes semantic similarity, is generated using *differences* in behavior.

Due to the diverse toolchains and architectures used and its closed source nature, binary analysis is particularly well suited to firmware. David et al. [20], created a static analysis tool to find CVEs in firmwares, and discovered hundreds of vulnerabilities. Feng et al. [27], took inspiration from image search research to find bugs in Internet of Things devices by converting CFGs into numerical vectors for similarity analysis.

## 9  Conclusion

We introduce IOVFI, a binary analysis framework that is architecture and compilation environment agnostic. Instead of measuring code properties, IOVFI abstracts functions into sets of input and output program states, information guaranteed to be stable across compilation environments. Our proof-of-concept implementation has a high .779 accuracy when identifying functions in binaries generated from various configurations, remains highly accurate even against purposefully obfuscated code, scales to large binaries, and generalizes to other architectures with minimal effort. It will be released as open source upon acceptance.

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
