# OpenReview forum: "Accurate Compiler, Optimization, and Architecture Independent Function Identification using Program State Transformations"
_JSYS/2022/Feb_Papers — Submitted to JSYS Feb 22_

### Official Review · Reviewer_wgkA · 2022-03-16
**Good work but needs more clarification**

**Decision:**

Weak accept: good paper with flaws that can be fixed in three months

**Review:**

Dear Authors, Overall, the work is nicely captured and experiments are comprehensive showing various benefits of IOVFI. However, the following aspects should be addressed in the paper.

- The authors motivate the paper for patching of executables but then go deep inside how to identify similarity between functions. Somewhere the main motivation is lost as the paper progresses forward. It might be useful to keep the main context into mind and explain how IOVFI can be used for patching of executable code or libraries. It might be useful to have some figure/flow to illustrate this. If the patching is not shown as a use case, then the abstract and introduction should be modified to bring the main problem of function identification into focus.

- Section 4.2 lists some limitations of the framework and Section 6 also focuses on the limitations. It would be good to club them together into Section 6. Similarly, Section 7 for future work is not needed and can be merged with Conclusion section together.

- In Sec. 4.3, N and Nbar are used to indicate number of functions and average number of functions per equivalence class. Usually, a bar is used to indicate some kind of negation or complementation. Using such a symbol here is confusing. May be authors should use some more appropriate notations.

- In Section 4.2, the authors mention "The IMF-SIM authors remained unresponsive". Such statements are not acceptable in formal writings. IMF-SIM authors are not obliged to respond. Instead, the authors could just say "The source code for IMF-SIM could not be obtained". In the same paragraph, it seems like a lot of unwanted content. "The BLEX authors supplied us their code, but it required significant engineering to execute with currently distributed Python modules. We invested two weeks of development and evaluation time." These kind of things are okay for a technical report but not for a paper. Two weeks is just nothing to get someone else's code working for you, when it was not even planned for open source. These things should just be removed.

- Figure 10 reports accuracy metric. There is no notion of %difference for accuracy. An improvement from 0.8 to 0.85 is much more useful than an improvement from 0.1 to 0.3, even though the former might be just 6% while the latter is 200%. I would suggest to just remove the last column from there. Similarly statements at many places are confusing - e.g., in abstract, "IOVFI achieves a 101% accuracy improvement over the most-recent BinDiff 6 ..." - How can accuracy increase by 101%? Typical understanding is that 100% is the highest accuracy. The suggestion is to remove accuracy improvement quantification in such manner from throughout the paper. If accuracy increases from 0.51 to 0.61, we should just say it as 10% (0.61-0.51) accuracy improvement rather than (0.61-0.51)/0.51 = 20%.

- Tables are also referred as figures in the paper. It is usually better to differentiate them as tables and figures. Figures 5-8, 10-13 are tables and not figures.

- The labels inside the boxes in Figure 14 are not readable easily for two reasons - the font size is small and the darker color hides dark fonts. Authors should correct this.

**Expertise:**

Follow the literature closely, last published 5+ years ago

**Useful:**

yes

---

### Official Review · Reviewer_sBYd · 2022-03-16
**Weak Accept**

**Decision:**

Weak accept: good paper with flaws that can be fixed in three months

**Review:**

This paper is proposing IOVFI, a decision tree-based method to identify functions in a binary code based on their behavior and input/output state vectors, which can be a great help in reverse engineering, malware detection, etc. Binary code analysis makes it more scalable than source code-based analysis. It has been successful in identifying functions in three common libraries in different OS versions.

The idea seems interesting, the paper is well-written, and the results are decent. There are some questions and issues:
- As a function's implementation changes (while the end result stays the same), how much of an impact could it have on the output of IOVFI? For example, would different sorting methods be classified as the same equivalence class, despite their differences in implementation?
- Functions using input pointers are quite common. I assume quite a few of the functions in the test binaries use them as well. In my opinion, this work would benefit from a method to analyze registers in the instructions (e.g., symbolic execution) (For example, if an input or a related register is found to be used as an address in a load instruction, the program would be able to allocate some memory and use its address as an input.)
- In my opinion, the part regarding the communication with other authors for their codes seems unnecessary.
- Minor writing issues
  - Some zeros (before the decimal point) are missing in the beginning of numbers between [-1, 1]
  - Some hyphens are missing (e.g., "function sized" --> function-sized?)

**Expertise:**

Published in this area in the last 5 years

**Useful:**

yes

---

### Official Review · Reviewer_tcfF · 2022-03-19
**Review for Accurate Compiler, Optimization and Architetcure Indenpendent Function Identifiaction Using Program State Transformations**

**Decision:**

Strong accept: excellent paper that will help the community

**Review:**


## Paper Summary
Semantic binary analysis is a key component for plagiarism detection, code debloating, security, and malware analysis. Because of the abstractions and non-readability, binary code is an attractive means that security engineers prefer for patching vulnerabilities. However, manually creating patches using existing semantic analysis techniques does not scale for large binaries. To address this issue, this paper aims at developing an automated solution for semantic analysis of binary code. While a number of automated solutions exist, they do not generalize well when the compilation environment changes even when different optimizations are applied. Therefore, this paper uses program state modifications as they are more stable as semantic function identifiers. The presented IOVec Function Identification (IOVFI) identifies the behavior of functions instead of the underlying code properties. IOVFI automatically discovers the program states and their behavior that is not affected when compilation environments are changed. When compared to the state-of-the-art approaches, IOVFI demonstrates significantly higher accuracy. The experimental results also demonstrate that IOVFI is more generalizable, scales to larger binaries, and is robust across different versions and architectures.
## Strengths

- The paper is nicely organized and written with care which makes it a really nice read. The assumptions and evaluation criteria are clearly stated.
- The experimental results support the key claims made in the paper. Particularly, the usage of program state changes for semantic identification is well bolstered by the excellent results found in the paper when compared to the state-of-the-art approaches.
- Solution is transferrable to a good degree across environments.
- Solution is scalable for large binaries with minimal additional effort.
- The impact of the work is highly significant
## Weaknesses
- The contribution itself feels very slim. The work uses the box mutational fuzzing tool to model for IOVec discovery.
- The greedy approach of discovering IOVecs could produce suboptimal results.
## Comments For the Authors
- The approach adopted for the derivation of pointers to generate high-quality IOVecs is really useful.
- I commend the authors for providing a thorough and comprehensive evaluation to test the effectiveness of IOVFI against the research questions mentioned in the paper. The case studies presented in section 5 clearly stand out in showing the efficacy of the approach.
- Some figures and tables can be readjusted for better positioning for better readability. (e.g., Figure 2 Figure 7).
- The toy example presented in Listing 1 is too simplistic. While it is clearly useful to understand the basic concepts in the paper, this does not clearly illustrate the complete details of the approach. A more detailed example would help position the approach better.
- Fuzzing time is allocated a 5 hours budget. The justification behind this selection was not clear. A sensitivity analysis can be presented to tackle this issue.
- The greedy nature of selecting IOVecs during the discovery phase may lead to suboptimality.
- Please mention the configurations at which the experiments were run.
- Can you please provide your comment on how the accuracy of IOVFI is affected under different configurations of the compiler?
- Can you add a few more sentences about Assumption 3. It was not clear why this was made and how not making this assumption IOVFI would fail.
- Is there any way to theoretically determine how the quality of IOVecs would affect the generalizability of the approach? Some empirical observation will also be useful that considers different threshold levels for code coverage and different fuzzing approaches.












**Expertise:**

Published in this area in the last 5 years

**Useful:**

yes

---

### Official Review · Reviewer_WRvA · 2022-03-29
**A solid work that still has some rooms for improvement!**

**Decision:**

Weak accept: good paper with flaws that can be fixed in three months

**Review:**

The paper presents a novel approach to binary semantic analysis based on a fascinating observation--that the program state modifications serve as a more stable semantic function identifier across different compilations, environments, and implementations. The paper proposes a framework that infers function semantics through program state changes. The paper also presents an implementation of the framework that leverages coverage-guided and mutational greybox fuzzing for inferring function semantics via program states and input structure layouts. The paper also evaluates the approach by showing the effectiveness of the approach on the GNU Coreutils package. Finally, the paper presents the results of 5 years of Ubuntu distributed versions function identification.

Semantic binary analysis has applications in diverse domains, including plagiarism detection, code debloating, and malware analysis. The key use case of binary semantic analysis is for composting systems, where systems are typically composed of integrating third-party and internal components. Without a source or an exact knowledge of how the library was generated, any user of a vulnerable library must either wait a long time for the developer to fix the library or rely on binary semantic analysis to identify and locate vulnerable functions.

Semantic binary analysis is a complex problem, and it has faced several challenges to be adopted in practice. So far, automated binary analysis measures binary code properties (e.g., order and type of instructions, memory locations accessed, or control flow) and approximates semantic similarity of functions based on code similarity. However, two programs might be semantically similar while they are far away in terms of code similarity. This paper leverages the stability of program state changes for functions across different compilations, environments, and implementations to address this challenge. The core idea is to observe and identify the behavior of functions instead of the underlying code, and then use the observed behavior as a unique function identifier. By observing data flow and program state transformations, IOVFI can classify functions, and, as a first-in-class feature, the IOVecs can transfer to different architectures with minimal effort.

I would like to first thank the authors for submitting their work to JSys!
There are several aspects of this work that I liked, and I would like to highlight them here:

1- The paper discusses the limitations of the proposed function identification when functions are dependent on system environment and execution context and functions that depend on the initial seed being set beforehand.

2- The paper presented case studies to demonstrate the effectiveness of the function identification approach. In particular, using program state changes for semantic similarity resulted in an approach that can perform function identification in obfuscated code similar to non-obfuscated code. In addition, evaluating the approach on libraries such as zlib, libpng, and libxml2 provides strong evidence that the approach would scale to large binaries.

There are several issues that I would like authors to either address or provide justification for why such analyses may not provide additional insights:

1- The approach uses a coverage-guided mutational fuzzer for discovering IOVecs. However, no experimental results indicate how increased program state change coverage resulted in better semantic differentiation in the function identification process.

2- Modern systems are typically highly configurable. In particular, specific assignment of values to configuration options (e.g, those define by #ifdef in c/c++) determine the execution path at runtime. Therefore, for accurate function identification, in addition to Input/Output, for this type of system, configuration information may also be needed for accurate function identification.

3- The proposed function matching approach adopted a strict policy for program state matching. The authors claimed that a different policy that more precisely compares program states could better account for inconsequential program state changes. I suggest the authors perform a sensitivity analysis to show how alternative approaches that may relax the condition by requiring a partial match between return values and allocated memory areas change the F-score.

**Expertise:**

Follow the literature closely, last published 5+ years ago

**Useful:**

yes